# The Use of Digital Portfolio in Higher Education before and during the COVID-19 Pandemic

**DOI:** 10.3390/ijerph182010904

**Published:** 2021-10-17

**Authors:** Soledad Domene-Martos, Margarita Rodríguez-Gallego, David Caldevilla-Domínguez, Almudena Barrientos-Báez

**Affiliations:** 1Department of Teaching and Educational Organization, Faculty of Education Sciences, University of Seville, 41013 Seville, Spain; margaguez@us.es; 2Department of Communication Theories and Analysis, Faculty of Information Sciences, Complutense University of Madrid, 28040 Madrid, Spain; davidcaldevilla@ccinf.ucm.es (D.C.-D.); almbarri@ucm.es (A.B.-B.)

**Keywords:** educational challenges, digital portfolio, online education and emerging technologies, higher education, COVID-19

## Abstract

This study is focused on the advantages and disadvantages of using a digital portfolio to improve the learning and evaluation processes in the initial teacher training of 4th-year students in the University of Seville (Spain). One of the interests of this research was to compare the learning capacities perceived by the students to improve their learning process before and during the COVID-19 pandemic. A qualitative, descriptive methodology was applied, identifying the most relevant dimensions, categories and codes for the analysis, management and interpretation of the opinions of the students, with a research triangulation (Cohen’s kappa coefficient) and a coding performed using the ATLAS.ti 8.4 software. The results show that the advantages with greater percentage correspond to the following categories: learning, usefulness of OneDrive, autonomy and evaluation. The greatest disadvantages detected were: time, uncertainty, usefulness of OneDrive and autonomy. There are differences in the perceptions of the students, between before and during the COVID-19 pandemic, about the learning capacities developed with the use of digital portfolio, since they consider that they have acquired more significant learning, greater self-regulation of their learning and greater reflection capacity.

## 1. Introduction

The perception of digital competence in education has expanded and evolved rapidly in recent years [1] and, therefore, its conception is not solid, as it is constantly changing. Considering that educational technology comprises a space for teaching, innovation and research, there is an imminent need to recognise it in the study plans of initial training, which will allow students to reconfigure their own learning process, since the circumstances demand it [2,3].

Currently, due to the situations derived from the COVID-19 pandemic, the education system has been forced to work with Information and Communication Technologies (ICT) and, thus, to provide an education of the best possible quality to facilitate and favour continuous, autonomous and versatile learning in adverse circumstances. The Spanish government has launched a programme to promote the technological transformation of education (known as Educa en Digital), due to this urgent need. However, it is worth highlighting that the proposition of the digital portfolio began before this lockdown situation, and such a strategy was used in the current pandemic context.

Being aware of this reality, we must consider all those instruments and resources that are at the service of education, including those which were not initially designed for teaching and learning purposes. This is the case of a digital portfolio that allows for creation of a place for identity construction [4], which is a tool focused on student evaluation, reflection and collaboration [5] that provides a transforming space of teaching–learning, modifying the involvement of the student [6].

In the stage of initial training, in higher education, the portfolio can be an excellent complement for the professional and academic development of the students, from both a methodological and an evaluative approach, in the didactic strategy of different subjects [7], allowing them to show what they learn, reflect on it, create a commitment [8] and establish renovated alternatives based on their professional development [9].

Some authors state that the original portfolio, regardless of whether it is digital or not, maintains some attributes that favour, particularly, the teaching–learning process, which was previously known as a rectangular and flat folder for carrying papers; nowadays, this has become a proposition that is used in the field of education as an effective and useful tool for both the teacher and the students. The portfolio has evolved in meaning, space and form, favouring the teaching–learning process, promoting those skills and capacities that are acquired in the process and strengthening the teacher–student bonds through feedback [10,11].

Regarding the particularities of this tool and what it can offer in the teaching–learning process, it could be stated that there are many possibilities for these two profiles, according to its essential functions in the formal scope. On the one hand, for the teacher, it is an evaluation method that justifies a marking system based on the continuity, process and progress of the students, as well as on the organisation and evidence of the product [12]. Thus, the evaluation process transcends mere quantification, through a qualitative assessment of the results [13], which allows evaluating other skills and competencies that can be developed with this instrument (creativity, autonomy, aptitude, metacognitive capacity, etc.), thus enabling the teaching practice to improve through feedback [10,14]. Moreover, the evaluation process through the portfolio offers opportunities for the exchange of knowledge experiences, that is, the possibility of creating and promoting innovating practices and a positive influence on teaching, learning and evaluation, by creating a new concept of the classroom as a place where students learn according to their individual pace, including the valuation of the reflective thinking, intuition and knowledge of each individual and the belief that difficulties can be overcome [15]. However, it provides the student with numerous possibilities and learning opportunities, since the realisation of this production is considered a process of knowledge generation, [16] self-evaluation [17] and development of metacognition (planning, self-evaluation and reflection) [18,19] and it improves the learning styles [20,21] and modifies the motivational beliefs about the subject [20]. Therefore, not only does it enrich the mere generation of knowledge [4], but its use also influences the personal development of the student [8,22]. Numerous studies highlight the benefits of digital portfolios in the learning process of students, thus confirming that the implementation of this tool produces a generalised satisfaction in those who use it as a teaching–learning method, motivated by the similarities of this methodology with the peculiarities of disruptive pedagogy [7], that is, the pedagogy that is favoured by the use of ICT, which in turn promotes the competencies linked to their use in future teachers [23]. From this approach, the good use of digital portfolios will guarantee the provision of capacities and skills that are strongly related to autonomy and freedom, which could determine the beginning of the change toward an innovative educational process in line with this era, thereby promoting the improvement of the quality of the university education system [24,25].

A large number of authors [8,10,16] conclude that the usefulness of this tool with respect to transversal capacities, skills and competencies promotes collaboration and teamwork, develops reflective practice, preserves identity, enables the use of other applications, regulates, controls and self-evaluates the learning acquired through feedback, and develops creativity due to the freedom upon the creation of the portfolio; to sum up, they consider that digital portfolios promote learning autonomy. This implies the development of professional competencies that students must acquire and the development of learning throughout life [26], transcending the mere educational experience and becoming a metacognitive experience that they can use to learn better and more thoroughly, in both specialised and personalised areas, favouring self-knowledge in the long term [15,27].

Furthermore, the strategic capacity of the portfolio expands with the benefits of the digital aspects, since it becomes a space where work is compiled in a dynamic manner and encourages the users to create and share the content and, thus, to collaborate. In addition, it allows storing the feedback and valuations, ensuring more agile and rich dynamics [28].

All these characteristics lead to the possibility of using this tool for the double modality of teaching (face-to-face and online), since its application is quite autonomous and active, and everything is recorded in a single document [29].

However, limitations have been found in the opinions of higher education students, who consider that the production of a digital portfolio poses additional work load and responsibility (which could generate certain stress levels) and requires a long time [7], connection to the Internet and the development of technological competencies [28,30].

Lastly, the circumstances could change at any time regarding the pandemic, which is why this study was focused on analysing the digital portfolio, determining its advantages and disadvantages based on the data gathered from 206 4th-year students of the Degree of Primary Education of the University of Seville (Spain). This experience was developed as a replication of a previous study [31]. We can stand out that, due to the characteristics of the students that constitute the sample, women will improve their learning skills in this pandemic period. The most original aspect of this study is the used of the digital portfolio online because of life circumstances.

## 2. Materials and Methods

Based on the characteristics of the study object and the objectives set, and from a holistic perspective, a qualitative descriptive methodology was selected [32,33] to interpret the opinions of the participating students. A content analysis of the students’ opinions on digital portfolios was performed, identifying those dimensions that were relevant for the analysis, management and interpretation of the comments. In turn, these were divided into categories, to which a specific code was assigned, with the aim of processing and coding the texts using the ATLAS.ti 8.4 software (ATLAS.ti, Berlin, Gernmany). This software allowed for creating a new project in which global relationships were established, in order to better understand and preserve the perceptions of the participants.

### 2.1. Objectives

The main objective of this study was to know the perceptions of the students toward the advantages and disadvantages of digital portfolios for the improvement of learning and evaluation in initial teacher training. To attain this goal, the following specific objectives were proposed: (O1) to carry out a content analysis to determine the perception of the students toward the learning and evaluative capacities developed in the digital portfolio; (O2) to verify the co-occurring capacities that impact such learning processes; (O3) to compare the learning capacities perceived by the students for the improvement of their learning process before and during the COVID-19 pandemic; and (O4) to establish improvement propositions for the learning of capacities and as an evaluation system through digital portfolio.

### 2.2. Sample

The sample was constituted by 206 4th-year students of the Degree of Primary Education of the University of Seville (Spain) from the academic years of 2018–2019 (*n* = 52), 2019–2020 (*n* = 79) and 2020–2021 (*n* = 75). In general, among their characteristics, 87.3% were women and 12.6% were men. The commonest age range was 20–22 years (67.9%), followed by 23–25 years (24.7%), 26–28 years (2.4%) and over 29 years (3.3%). The academic access to the university degree was mostly through baccalaureate (82.5%), followed by the Degree of Early Childhood Education (11.1%). With these data, we can conclude that the general profile of the participants is that of a 20–22-year-old woman who has entered the university through the baccalaureate.

### 2.3. Instrument

A digital portfolio was used to know the perceptions of the students toward the advantages and disadvantages of its use. In the academic year 2018–2019, the subject was taught face-to-face, whereas in the academic years 2019–2020 and 2020–2021 the subject was taught online due to the COVID-19 pandemic.

The digital portfolio was carried out using the Office 365 platform of the University of Seville, which consists of a set of communication and collaboration tools based in the Cloud and which integrate all those applications of Microsoft Office (Microsoft: Redmond, WA, USA) free of charge. Through the Cloud, the students were able to interact and receive feedback from their peers and faculty members. The basic structure of the digital portfolio was made of a presentation of the student, a portfolio index and the content of it. The content consisted of information written by the students. They should be able to write about what have they seen, worked or learned in class, what are the things that they would have to do for the next day, explain what are the steps they need to follow in each activity. Also, they should include in their digital portfolio their class notes and all the books, magazines and webs used in class. Each student will we able to choose the format of the presentation of the digital portfolio.

### 2.4. Data Analysis Procedure

The analysis of the data consisted of a thorough reading of the digital portfolios to create a system of categories that represented the perceptions of the students regarding the study object. As a result of this process, a system of dimensions (advantages and disadvantages) was established around two major foci of interest proposed in the objectives of this study, as well as response categories and codes. In the creation of categories for the two major dimensions, five common categories were established: autonomy, responsibility, evaluation, learning and usefulness. However, within the indicator of advantages, the innovation category was included, due to its high content of evidence. On the other hand, the disadvantages included: time, uncertainty and none for the same reason. Categories were created in every dimension. See Table 1.

In the analysis, the code-document cross-tabulation was performed, enabling the extraction of the absolute frequency of each code, which, in this case, corresponded to the academic year and percentages. Evidence maps of the students are presented, to show the maximum and minimum load of the categories.

To ensure the validity and reliability of the obtained results, a three-coder triangulation was conducted, related to the content analysis and based on the system of dimensions, categories and codes. The consistency level among them was calculated with Cohen’s kappa coefficient. The validity and reliability were guaranteed by the high consistency index obtained among the coders [34]. The rounded value of the kappa coefficient was 0.99. The probability of coincidences between the codifications due to chance was discarded (*p* = 0.000).

We also analysed the co-occurrence of the categories with greater and smaller load, with the aim of establishing a pattern of association about the learning and evaluation capacities developed with the digital portfolio. To clarify the co-occurrence between categories, the most representative evidence maps are presented.

Lastly, we compared the learning capacities perceived by the students for the improvement of their learning process before and during the COVID-19 pandemic, in order to know how the learning modalities (face-to-face or online) affected the development of the subject.

New network entities corresponding to the semantic maps or networks were created, by linking the groups of codes and aggregating common neighbours or codes, according to the interest of the interpretation of the data. Moreover, for some general networks, we included the view of comments and frequency of the codes (corresponding to R, which is the number of times that the code was fixed in the text, and D, which is the density of the links).

To include evidence of a specific code, the same procedure was carried out, although, instead of aggregating common neighbours or codes, neighbours of verbatim quotes were aggregated.

## 3. Results

According to the objectives of the study, Table 2 presents the data of the perceptions of the students toward the advantages and disadvantages of using the digital portfolios.

As can be observed, the advantages with the highest percentages correspond to the following categories: learning, usefulness of OneDrive, autonomy and evaluation. On the other hand, the disadvantages with the highest percentages corresponded to the following categories: time, uncertainty, usefulness of OneDrive and autonomy. There are categories without any disadvantages, such as innovation and accessibility, as well as categories without any advantages, such as time and uncertainty.

Figure 1 shows the advantages of using the digital portfolio according to the students, selecting those with the highest and lowest loads.

The students valued the control and pace of their learning positively, although to a lesser extent compared to the development of digital competence, possibly due to the fact that they are used to employing different digital tools. They also had a good opinion about the fact that the digital portfolio allows them to organise and safeguard the material of the subject, without considering the characteristic of sustainable resource. Regarding autonomy, they estimated the possibility provided by the portfolio to demonstrate their learnings with their own style, although they did not consider the opportunity to express their ideas autonomously. They also had a positive view on the novelty of the digital portfolio as an alternative evaluation system. For some students, it seemed important to establish evaluation criteria from the beginning of the subject. Innovation in the development of the portfolio was expressed in opinions related to the promotion of creativity and originality, although with a lower load than the possibility of applying ICT for its design, structure and format.

Regarding the disadvantages (Figure 2), the perceptions of the students were mainly focused on the “time” category, considering that the development of the digital portfolio requires a long time and highlighting the duration of the evaluation process by the teacher as a disadvantage. This implies that the evaluation required a longer time compared to previous evaluation systems. Moreover, the participants were concerned about the uncertainty of gathering all the information of the subject in the portfolio without previously knowing whether the selected information is correct, and they pointed out that, in some cases, they did not receive enough feedback from the faculty members. The complexity in the use of OneDrive was another disadvantage perceived by the participants, although the fact that the digital portfolio required access to the Internet and to a computer was valued as a disadvantage with a lower load. Autonomy, specifically the organisation of the portfolio at the beginning of the subject, was valued as an important disadvantage, and very few students mentioned the lack of honesty in the presentation of their work for being carried out outside of the classroom.

The co-occurrence of codes with respect to the advantages (Table 3) is mainly observed in the positive valuation of the digital portfolio for the self-regulation of learning (VPA4), allowing the students to better acquire the knowledge of the subject (VPA2) and thus producing meaningful learning (VPA1). This is reflected in their opinions: “Revising the documents from the classes every day, every week and at the end of the lectures helped me to consolidate the knowledge” (1:68) and “Thanks to this portfolio, I followed the subject day by day and I worked on it every day, learning every day and achieving a better assimilation of the contents, in such a way that, when the day of the exam approached, I did not have to study everything anew, but I merely had to revise everything that I had learned” (1:24). Similarly, the portfolio encouraged them to reflect (VPA5): “Moreover, it also helps the students to organise their knowledge and reflect on what they are learning and what they have learned” (3:95).

Regarding the usefulness of OneDrive, the participants highlighted the versatility of the tool (VPU1), as it allows organising and safeguarding the material of the subject (VPU9): “The portfolios are performed intuitively, and it helps the students to order, classify and modify the relevant documents” (3:135). The students declared that autonomy in the organisation of the material (VPAU5) helped them to self-regulate their learning (VPA4): “Organising our own knowledge and learning, and carrying out a continuous and constant organisation and planning of the information that we had to study; all done by the students themselves in their own personal style” (1:281).

Furthermore, according to the participants, the development and use of the digital portfolio promotes self-evaluation (VPE5), since the student controls his/her learning (VPA4): “It promotes self-evaluation, control and participation of our learning” (2:335); “The students can evaluate themselves and control their learning” (2:69); “It facilitates a constant working habit, and it is also a good evaluation tool that allows the student to reflect on and evaluate his/her own learning” (1:92).

Regarding the co-occurrence of codes in the disadvantages (See Table 4), the time required to evaluate the portfolio and its follow-up by the teacher (IPT2) throughout the trimester was associated by the participants with poor feedback, which was bidirectional in some cases: teacher–student and student–teacher (IPT1). This code seems to be related to the poor validity of the results (IPE1): “It takes a long time for the students and the teacher to carry out and evaluate the digital portfolio, respectively” (2:212); “Both students and teachers have to dedicate too much time” (2:115). It also appears to be related to the uncertainty of the unknown (IPT2): “I found the main disadvantages at the beginning of the creation of my portfolio, since I was a bit lost due to the fact that I had never worked with this type of tool. I did not know what I had to write down or how I ended up with a lack of certain information during the first days, as I tried to copy everything word by word, and so I missed many important things” (1:15).

Table 5 presents the data obtained regarding the advantages and disadvantages perceived by the students before and during the COVID-19 pandemic.

The advantages for the academic year 18/19 regarding learning (VPA) obtained a higher percentage compared to the COVID-19 pandemic years: meaningful learning (VPA1), self-regulation of learning (VPA4) and reflection (VPA5). With respect to autonomy, the percentage of advantages before and during the pandemic is not considerable in motivation and self-confidence (VPAU2), autonomous and integral work (VPAU3) and in student participation.

With regard to innovation (VPI), there is a considerable difference between the percentages before and during the pandemic, with the use of space and time (VPI1) being more advantageous during the pandemic years. Similar results were obtained for the use of ICT: carrying out the portfolio (VPI2). Responsibility (VPR1) was considered by the participants with a higher percentage in face-to-face education versus online education. With regard to the usefulness of OneDrive (UPU), the students recognised the versatility of the tool (UPU1) as an advantage with a higher percentage during the pandemic. They also recognised it for use (IPU2) in online work (See Figure 3).

With respect to the disadvantages (see Figure 4), there are differences in learning (IPA), with no disadvantages before the pandemic and some disadvantages during the pandemic; in this case, the acquisition of meaningful learning (IPA1) was considered a disadvantage. In autonomy (IPAU), the percentage of disadvantages was higher before the pandemic: dishonesty in carrying out the portfolio (IPAU1), not knowing how to organise oneself at the beginning (IPAU2) and the lack of group coordination (IPAU3). Uncertainty (IPI) obtained a higher percentage as a disadvantage before the pandemic: feedback (IPI1), fear of the unknown (IPI2) and being excessively alert about gathering information in class (IPI3). Time (IP) was also a disadvantage with a higher percentage before the pandemic: the time required from the students and daily effort (IPT1) obtained a lower percentage during the pandemic. The percentage of disadvantages of the usefulness of OneDrive (IPU) regarding the complexity of use of the tool (IPU1) and the required access to the Internet and to a computer (IPU2) was much higher in the pandemic years.

## 4. Conclusions

With respect to the first objective of this study, we can conclude that the advantages perceived by the participants in the three academic years overcome the disadvantages, with the former being almost two thirds of all the opinions provided. As it happened in other investigations, the students that have participated in this study had a positive opinion about the development of the produced learning [10,11,14], the usefulness of OneDrive [29], autonomy [25,26] and evaluation in the use of the digital portfolio [12,13]. In the same way, they considered that their learning process was more self-regulated, with greater control and learning pace in the tasks performed for the attainment of the objectives set in the subject, and also with a more effective studying of the subject matter. In relation to the usefulness of OneDrive, they highlighted the ease for organising and safeguarding the material to have all the information of the subject more ordered, which is in line with the autonomy dimension, as it favours the organisation of the material. Moreover, the participants considered the digital portfolio as an alternative and more positive evaluation system compared to the traditional system. These results are in agreement with those of previous studies conducted in the field of health [35,36,37]. However, the participants had a negative opinion about the time required to carry out the portfolio [7,30,38] and the uncertainty [28] caused by this new tool. Some students considered learning a new application like OneDrive and the degree of autonomy as disadvantages, since their learning and evaluation process had been, to this point, more directed, and they were somewhat lost at the beginning in terms of how to organise the material. Therefore, the 4th-year students perceived that the use of the digital portfolio had more advantages than disadvantages for the learning of the subject and as an evaluation system in the three academic years analysed.

For the second objective, after interpreting the co-occurrence analysis, we can conclude that the capacities perceived by the students of this subject as advantages with the greatest impact on the learning processes are: self-regulation of learning [18], deeper understanding of the subject [16] and, consequently, acquisition of more meaningful learnings [17,19]. The students considered that they performed the proposed tasks in a better way, established their own goals and evaluated their self-efficacy to keep learning; they also perceived the generation of a good environment in the large group and in smaller working groups. In relation to the usefulness of OneDrive, they considered that it is a very versatile tool, as it allows organising and safeguarding the material [29]. Regarding the disadvantages in the co-occurrence analysis, time was the greatest, followed by the follow-up performed by the teacher, which was associated with the poor feedback received. Other investigations have achieved the same results [7,28,30]. The relationship established by the students of the sample in these categories can be due to the amount of time spent to carry out the portfolio and the poor feedback, which, in some cases, was bidirectional (teacher–student and student–teacher); consequently, they did not recognise the validity of the contributions of their digital portfolio. These data are in contrast with those of previous studies [38,39] on student portfolios, which reported that one of the most valued characteristics was the feedback received by the teacher.

For the third objective, (i.e., to compare the learning capacities perceived before and during the COVID-19 pandemic), it was detected that the use of the digital portfolio was more advantageous for the attainment of more meaningful learning, greater self-regulation of learning and reflection during the face-to-face academic year. However, there were no changes in the valuation of student autonomy. This could be due to the fact that carrying out the digital portfolio requires organisation in both education modes (face-to-face and online) [40]. With respect to the innovation category, there was a higher percentage of advantages during the pandemic, which could be due to the fact that the students had to reinvent themselves often to learn the subjects in the absence of the teacher. Moreover, the use of space and time during the pandemic had more advantages than before the pandemic, since all teaching was online, thus the students did not have to carry out the portfolio at a certain time or place. The responsibility category obtained a better opinion before the pandemic, probably due to the confidence of having the teacher in the classroom to solve doubts. In the comments about the usefulness of OneDrive, the participants recognised a higher percentage in versatility and the use of this tool during the pandemic. This aspect could be due to the fact that they experienced the use of this application in online education. For the disadvantages, the learning category did not obtain any disadvantage before the pandemic, although the autonomy category obtained a higher percentage in face-to-face education [35]. This was probably due to the much lower interaction and cooperation between students during the pandemic, which implied fewer possibilities of sharing or copying. Furthermore, in the pandemic years, the students were given a tutorial on the use of OneDrive and how to carry out the digital portfolio; before the pandemic, the instructions were provided in face-to-face sessions. Uncertainty as feedback, fear of the unknown and being excessively alert about gathering all the information obtained a higher percentage before the pandemic. This could be due to the fact that, during the pandemic, the subject was better panned, providing more detailed instructions and presentations compared to face-to-face education [35]. The attention of the students at home was not affected by any external element; however, in the classroom, distraction can be much greater and hard to control. Time was also a disadvantage before the pandemic; the daily time and effort of the student obtained a lower percentage during the pandemic, probably due to the online character of education in this period, which allowed the students to make a better use of their time at home to focus on carrying out the digital portfolio. The percentage of disadvantages of the usefulness of OneDrive regarding the complexity of the use of the tool and the required access to the Internet and to a computer was much greater in the pandemic years, since it was the only tool to be used in this online period, without the possibility of obtaining direct information about its functioning.

To conclude, we present some propositions to improve the use of digital portfolios in higher education by investigating other topics and directions of future research: to focus on the consistency between the evaluation criteria and their application in the digital portfolio, making use of the management of technological skills; teachers must not delay the evaluation of the digital portfolio; to provide guidelines in all the class sessions to avoid the uncertainty caused by having to gather the information; and teachers should give constant feedback, to ensure, at all times, that the students can know the progress of their work before obtaining the final mark. The limitations of the investigation are the sample, because it can be more extensive, and also the method used. We could use a quantitative method in order to obtain better results.

## Figures and Tables

**Figure 1 ijerph-18-10904-f001:**
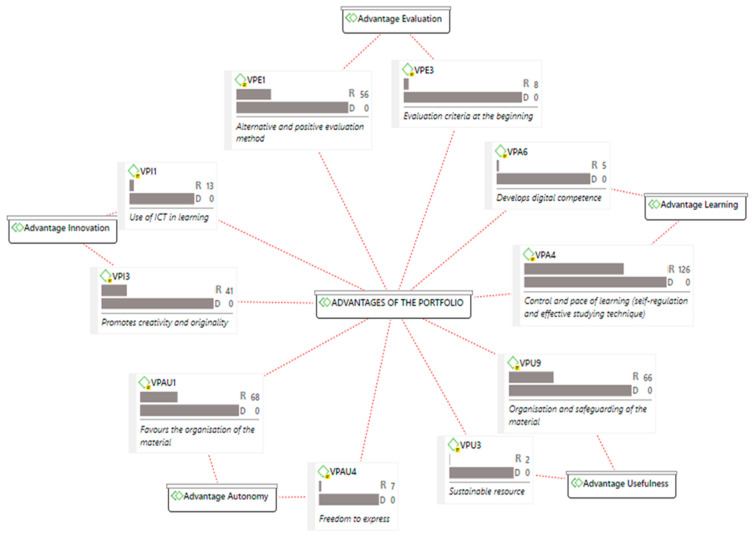
Map of maximum and minimum advantages by category/code (R: rooting; D: density).

**Figure 2 ijerph-18-10904-f002:**
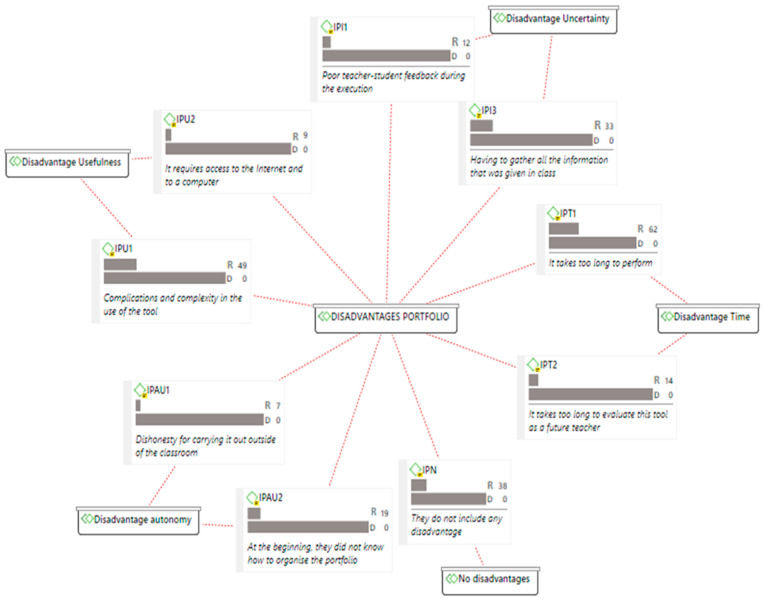
Map of maximum and minimum disadvantages by category/code (R: rooting; D: density).

**Figure 3 ijerph-18-10904-f003:**
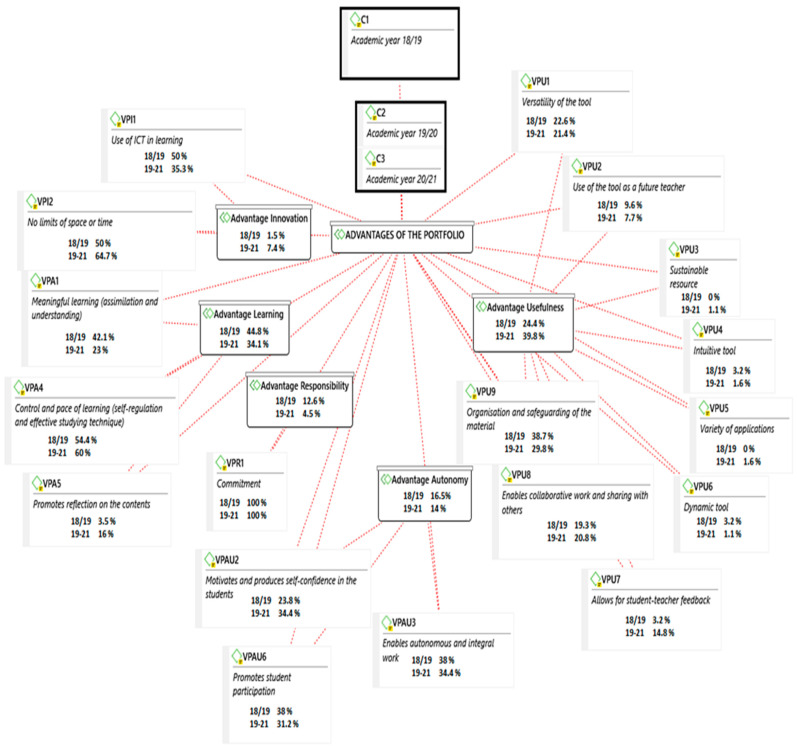
Advantages of the digital portfolio before and during the COVID-19 pandemic.

**Figure 4 ijerph-18-10904-f004:**
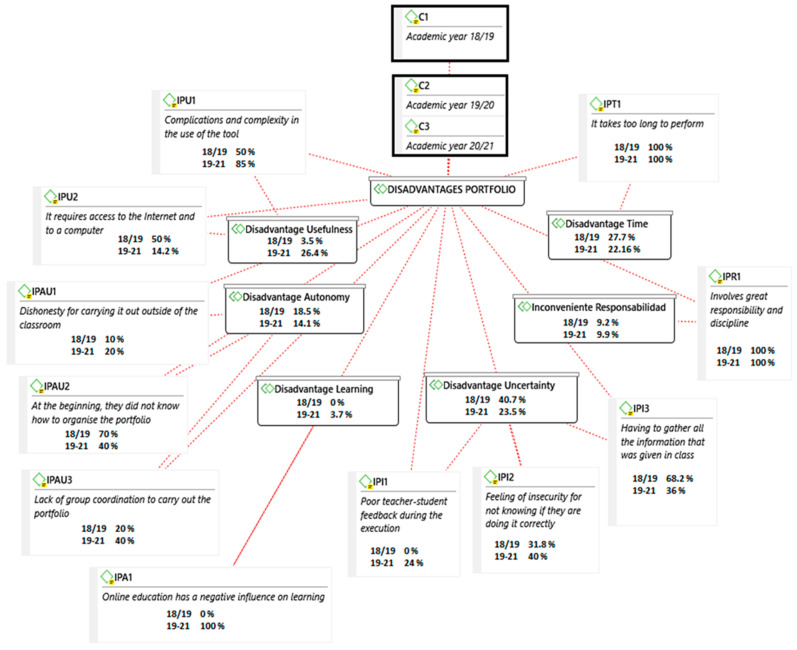
Disadvantages of the digital portfolio before and during the COVID-19 pandemic.

**Table 1 ijerph-18-10904-t001:** Analysis of the advantages and disadvantages of the portfolio according to the perceptions of the participants.

CATEGORIES	CODES	VALUATION	DEFINITION
**Accessibility**	VPAC	Advantages	Use in all educational stages (VPAC1)
**Learning**	VPA	Advantages	Significant learning (VPA1), in-depth knowledge (VPA2), research and curiosity (VPA3), self-regulation (VPA4), reflection (VPA5), digital competence (VPA6)
	IPA	Disadvantages	Incomplete learning due to a lack of face-to-face interaction (IPA1)
**Autonomy**	VPAU	Advantages	Organisation of the material (VPAU1), motivation and self-confidence (VPAU2), autonomous and integral work (VPAU3), freedom of expression (VPAU4), interests and skills of the students (VPAU5), participation (VPAU6)
	IPAU	Disadvantages	Dishonesty in the development of the portfolio (IPAU1), not knowing how to organise one’s own work at the beginning (IPAU2), lack of group coordination (IPAU3)
**Evaluation**	VPE	Advantages	Positive and alternative method (VPE1), more learning evidence (VPE2), evaluation criteria at the beginning (VPE3), skills and abilities (VPE4), self-evaluation (VPE5), individual and group work (VPE6)
	IPE	Disadvantages	Poor validity of the results (IPE1)
**Uncertainty**	IPI	Disadvantages	Poor teacher-student feedback (IPI1), the realisation is correct or not (IPI2), gathering as much information as possible from the lectures is stressful (IPI3)
**Innovation**	VPI	Advantages	Use of ICT (VPI1), no limit of space or time (VPI2), creativity and originality (VPI3)
**Responsibility**	VPR	Advantages	Commitment and responsibility (VPR1)
	IPR	Disadvantages	Responsibility and discipline (IPR1)
**Time**	IPT	Disadvantages	The students have to dedicate a long time (IPT1), the evaluation takes a long time from the teacher (IPT2), hard and tedious work (IPT3)
**Usefulness OneDrive**	VPU	Advantages	Versatility (VPU1), use as future teacher (VPU2), sustainable resource (VPU3), intuitive tool (VPU4), variety of applications (VPU5), dynamic tool (VPU6), feedback (VPU7), collaborative work (VPU8), organisation and safekeeping of the material (VPU9)
	IPU	Disadvantages	Complex use (IPU1), it requires access to the Internet and to a computer (IPU2)

VPAC (Accessibility Advantages), VPA (Learning Advantages), IPA (Learning Disadvantages), VPAU (Autonomy Advantages), IPAU (Autonomy Disadvantages), VPE (Evaluation Advantages), IPE (Evaluation Disadvantages), IPI (Uncertainty Disadvantages), VPI (Innovation Advantages), VPR (Responsibility Advantages), IPR (Responsibility Disadvantages), IPT (Time Disadvantages), VPU (Usefulness OneDrive Advantages), IPU (Usefulness OneDrive Disadvantages).

**Table 2 ijerph-18-10904-t002:** Categories, codes and frequencies of the opinions of the students about the digital portfolio.

CATEGORIES	CODE	FREQUENCIES	%
Accesibility	VPAC	8	0.6
Learning	VPA	246	19.5
	IPA	8	0.6
Autonomy	VPAU	188	14.9
	IPAU	40	3.2
Evaluation	VPE	154	12.2
	IPE	23	1.8
Uncertainty	IPI	72	5.7
Innovation	VPI	77	6.1
Responsability	VPR	37	2.9
	IPR	26	2.1
Time	IPT	110	8.7
Usefulness of OneDrive	VPU	213	16.9
	IPU	58	4.6
TOTAL	VP (Advantages) = 923/73.2%		
	IP (Disadvantages) = 337/26.7%		
	1260		

**Table 3 ijerph-18-10904-t003:** Co-occurrence of the opinions of the students with respect to the advantages of the digital portfolio.

	VPA1	VPA2	VPA4	VPA5	VPAU1	VPAU3	VPAU5	VPAU6	VPE1	VPE2	VPE5	VPI3	VPU1	VPU7	VPU8	VPU9
**VPA1**		5	13	3	2	3		1	8						3	3
**VPA2**	5		1			1	1									1
**VPA4**	13	1		7	14	4	2	7	4	6	7	5				
**VPA5**	3		7					1	1							
**VPAU1**	2		14			2	5				1	3	4			6
**VPAU3**	3	1	4		2		5	1	1			3	1			1
**VPAU5**		1	2		5	5						2				1
**VPAU6**	1		7	1		1			1	1						
**VPE1**	8		4	1		1		1		6				1	1	
**VPE2**			6					1	6		1					1
**VPE5**			7		1					1						
**VPI3**			5		3	3	2									
**VPU1**					4	1									3	7
**VPU7**									1						6	
**VPU8**	3								1				3	6		1
**VPU9**	3	1			6	1	1			1			7		1	

**Table 4 ijerph-18-10904-t004:** Co-occurrence of the opinions of the students regarding the disadvantages of the digital portfolio.

CO-OCCURRENCE OF DISADVANTAGES	IPE1	IPI2	IPI3	RIP1	IPT1	IPT2	IPT3	IPU2
**IPT1**	2	1	1	1		8	2	1
**IPT2**	1				8			2
**IPT3**	1				2			1

**Table 5 ijerph-18-10904-t005:** Categories, codes and percentages of the opinions of the students about the digital portfolio before and during the COVID-19 pandemic.

CATEGORIES	CODES	18/19	19/20–20/21
**ADVANTAGES**			
		Percentage	Percentage
**Learning**	VPA1|VPA4|VPA5	44.8%	34.1%
**Autonomy**	VPAU2|VPAU3|VPAU6	16.5%	14%
**Innovation**	VPI1|VPI2	1.5%	7.5%
**Responsibility**	VPR1	12.6%	4.5%
**Usefulness**	VPU1-VPU9	24.4%	39.8%
**DISADVANTAGES**			
**Learning**	IPA1	0	3.4%
**Autonomy**	IPAU1|IPAU2|IPAU3	16.7%	13.8%
**Uncertainty**	IPI1|IPI2|IPI3	36.6%	21.2%
**Time**	IPT1	25%	19.0%
**Responsibility**	IPR1	8.3%	8.9%
**Usefulness**	UPU1|IPU2	3.3%	23.7%
**None**	IPN	10%	13.5%

## Data Availability

Not applicable.

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
