# Peer review of "The Use of Digital Portfolio in Higher Education before and during the COVID-19 Pandemic"

_ijerph, 2021, doi:10.3390/ijerph182010904_

Round 1
Reviewer 1 Report
This paper mainly describes the advantages and disadvantages of using digital portfolio to improve the learning and evaluation processes in the initial teacher training of 4th-year students in the University of Seville (Spain) During the COVID-19 pandemic.
The topic is interesting. If the some following key points could be explained would be better.
(1) what is the new point or original points compared to other paper about this topic? It is better to give more details about this in the first section or introduction section. It is difficult for me to gain the original point in this paper.
(2) It is better to add more content related to the scope: Women's Health of the this journal. It seems there is no part in this paper to talk about this. It is better to analysis the experiment result or do experiment about from women's health viewpoints during the COVID-19 pandemic
(3) Give more details like image about what kind of digital portfolio are used during the teacher training of 4th-year students. It is better to explain more about digital portfolio.
(4) If possible, gain feedback or comments from students about using digital portfolio through the survey or interview to students.
Author Response
Response to Reviewer 1 Comments:
Point 1: what is the new point or original points compared to other paper about this topic? It is better to give more details about this in the first section or introduction section. It is difficult for me to gain the original point in this paper.
Response 1: We have included this point in end of the introduction. The most original aspect of this study is the used of the digital portfolio online because of the pandemic circumstances.
Point 2: It is better to add more content related to the scope: Women's Health of the this journal. It seems there is no part in this paper to talk about this. It is better to analysis the experiment result or do experiment about from women's health viewpoints during the COVID-19 pandemic
Response 2: We have included this point in end of the introduction. We can stand out that, due to the characteristics of the students that constitute the sample, women will improve their learning skills in this pandemic period.
Point 3: Give more details like image about what kind of digital portfolio are used during the teacher training of 4th-year students. It is better to explain more about digital portfolio.
Response 3: We have included this point in the paragraph of Instrument
Point 4: If possible, gain feedback or comments from students about using digital portfolio through the survey or interview to students.
Response 4: In the paragraph of Results we have included the comments of the students. As an example:
This is reflected in their opinions: “Revising the documents from the classes every day, every week and at the end of the lectures helped me to consolidate the knowledge” (1:68) and “Thanks to this portfolio, I followed the subject day by day and I worked on it every day, learning every day and achieving a better assimilation of the contents, in such a way that, when the day of the exam approached, I did not have to study everything anew, but I merely had to revise everything that I had learned” (1:24). Similarly, the portfolio encouraged them to reflect (VPA5): “Moreover, it also helps the students to organise their knowledge and reflect on what they are learning and what they have learned” (3:95).
Regarding the usefulness of OneDrive, the participants highlighted the versatility of the tool (VPU1), as it allows organising and safeguarding the material of the subject (VPU9): “The portfolios are performed intuitively, and it helps the students to order, classify and modify the relevant documents” (3:135). The students declared that autonomy in the organisation of the material (VPAU5) helped them to self-regulate their learning (VPA4): “Organising our own knowledge and learning, and carrying out a continuous and constant organisation and planning of the information that we had to study; all done by the students themselves in their own personal style” (1:281).
Furthermore, according to the participants, the development and use of the digital portfolio promotes self-evaluation (VPE5), since the student controls his/her learning (VPA4): “It promotes self-evaluation, control and participation of our learning” (2:335); “The students can evaluate themselves and control their learning” (2:69); “It facilitates a constant working habit, and it is also a good evaluation tool that allows the student to reflect on and evaluate his/her own learning” (1:92).
Reviewer 2 Report
Broad comments:
It is generally a well written paper. The subject of the paper is important nowadays, as the learning system changed in the time of Covid-19 pandemic. The Authors compared the learning capacities perceived by the students to improve their learning process before and during the COVID-19 pandemic. It was found out that the advantages perceived by the participants in the three academic years overcome the disadvantages. The analysis revealed that the capacities perceived as advantages with the greatest impact on the learning processes of the students are: self-regulation of learning, deeper understanding of the subject and, consequently, acquisition of more meaningful learnings. The study revealed that there are differences in the perceptions of the students, between before and during the COVID-19 pandemic, about the learning capacities developed with the use of digital portfolio, since they consider that they have acquired a more significant learning, greater self-regulation of their learning and greater reflection capacity.
The subject of the paper is important nowadays, as the learning system changed in the time of Covid-19 pandemic. The main strengths of the paper are: i) clear problem statement, ii) very good literature study covering the most important publications iii) methodological correctness, iv) topicality and relevance of the subject taken. The main weaknesses of the paper are: i) group references in the Introduction, ii) lack of references in the text to several Tables and Figures,; iii) lack of clear information about the contribution to the body of knowledge of this paper (please see Specific comments below), limitations of this work and directions of the future research. Therefore I recommend paper acceptance after minor revision.
Specific comments:
INTRODUCTION
Lines 59 and 66 – Please avoid group references, e.g. [4-6], [7-9]. It is advised to clearly state what the contribution of each work was.
- Materials and methods
There is no reference to table 1 in the text. Please add.
- Results
There is no reference to table 4, Figure 4 and Figure 5 in the text. Please add.
Figures 1 and 2 are difficult to read. Please improve their quality.
- Discussion and conclusions
There is no information about the limitations of this work and directions of future research. Please add the missing information.
Lines 403-405: “The students had a positive opinion about the development of the produced learning [10,11,14], the usefulness of OneDrive [29], autonomy [25,26] and evaluation in the use of the digital portfolio [12,13]”, Lines 417-418: “However, they had a negative opinion about the time required to carry out the portfolio [7,30,38] and the uncertainty [28] caused by this new tool”
Lines 426-430: “For the second objective, after interpreting the co-occurrence analysis, we can conclude that the capacities perceived as advantages with the greatest impact on the learning processes of the students are: self-regulation of learning, deeper understanding of the subject and, consequently, acquisition of more meaningful learnings [16–19]”.
Lines 437-439: “Regarding the disadvantages in the co-occurrence analysis, time was the greatest, followed by the follow-up performed by the teacher, which was associated with the poor feedback received [7,28,30]”:
Discussion should concern the research which was carried out. In Discussion section there are many references to other authors’ works, therefore it is not clear if the results which are presented in this paragraph are result of this research (based on Spanish students opinion) or results of the literature review. Please clearly write what is the contribution of this paper compared to the works of other authors.
Author Response
Point 1: INTRODUCTION. Lines 59 and 66 – Please avoid group references, e.g. [4-6], [7-9]. It is advised to clearly state what the contribution of each work was.
Response 1: We have changed the references.
Point 2: Materials and methods. There is no reference to table 1 in the text. Please add.
Response 2: We have included the reference to the table number 1
Point 3: Results. There is no reference to table 4, Figure 4 and Figure 5 in the text. Please add. Figures 1 and 2 are difficult to read. Please improve their quality.
Response 3: We have included in the text the reference to the table number 4 and the figure number 3 and 4. There is no figure number 5. We have improved the quality of the figure number 1 and 2.
Point 4: Discussion and conclusions. There is no information about the limitations of this work and directions of future research. Please add the missing information.
Response 4: To conclude, we present some propositions to improve the use of digital portfolio in higher education by investigating other topics and directions of future research: to focus on the consistency between the evaluation criteria and their application in the digital portfolio, making use of the management of technological skills; teachers must not delay the evaluation of the digital portfolio; to provide guidelines in all the class sessions to avoid the uncertainty caused by having to gather the information; and teachers should give constant feedback, to ensure, at all times, that the students can know the progress of their work before obtaining the final mark. The limitations of the investigation are the sample, because it can be more extensive, and also the method used. We could use a quantitative method in order to obtain better results.
Lines 403-405: “The students had a positive opinion about the development of the produced learning [10,11,14], the usefulness of OneDrive [29], autonomy [25,26] and evaluation in the use of the digital portfolio [12,13]”, Lines 417-418: “However, they had a negative opinion about the time required to carry out the portfolio [7,30,38] and the uncertainty [28] caused by this new tool”
We have not made any changes because the three investigations conclude the same way in this subject.
Lines 426-430: “For the second objective, after interpreting the co-occurrence analysis, we can conclude that the capacities perceived as advantages with the greatest impact on the learning processes of the students are: self-regulation of learning, deeper understanding of the subject and, consequently, acquisition of more meaningful learnings [16–19]”.
Revised. See the control changes.
Lines 437-439: “Regarding the disadvantages in the co-occurrence analysis, time was the greatest, followed by the follow-up performed by the teacher, which was associated with the poor feedback received [7,28,30]”:
We have not made any changes because the three investigations conclude the same way in this subject
Point 5: Discussion should concern the research which was carried out. In Discussion section there are many references to other authors’ works, therefore it is not clear if the results which are presented in this paragraph are result of this research (based on Spanish students opinion) or results of the literature review. Please clearly write what is the contribution of this paper compared to the works of other authors.
Response 5: Revised. See the control changes.
Reviewer 3 Report
I'm reading the article "The use of the digital portfolio in higher education before and during the COVID-19 pandemic" sent to ijerph. I congratulate the authors for the article and for researching such an important area.
The article starts off very well - the introduction clearly describes the problem, what is proposed and how it is proposed. It seems to me all very well until the results.
The tables are very unfriendly and important information is not taken - and there are problems like table 3 which doesn't even have a header or table 4 which is a mess. The figures are very difficult to read and useless for the reader.
The discussion seems short and unexciting to anyone who reads it - it seems that the authors are in a hurry.
What I propose is to give the article to someone who is not familiar with the investigation to read. Is this person able to get information from tables and figures? I'm sorry because the article is interesting, but the authors aren't writing for the reader - they're just complying. I propose that they talk to someone who will help them transform the (poor) tables and futures and transform them into tables and figures that are at the level of text and research.
Author Response
Point 1: The tables are very unfriendly and important information is not taken - and there are problems like table 3 which doesn't even have a header or table 4 which is a mess. The figures are very difficult to read and useless for the reader.
Response 1: Tables number 3 and 4 have a title in the manuscript. Tables have been reviewed in order to improve the quality for the reader.
Point 2: The discussion seems short and unexciting to anyone who reads it - it seems that the authors are in a hurry.
Response 2: We have improve the discussion with the instructions of the reviewers.
Point 3: What I propose is to give the article to someone who is not familiar with the investigation to read. Is this person able to get information from tables and figures? I'm sorry because the article is interesting, but the authors aren't writing for the reader - they're just complying. I propose that they talk to someone who will help them transform the (poor) tables and futures and transform them into tables and figures that are at the level of text and research.
Response 3: We have reviewed tables and figures keeping the most relevant information considering that potential readers work in this subject and have knowledge about it.
Round 2
Reviewer 1 Report
thanks for your comments about my question!
now it is clear for me